# Counterexample-Guided Learning of Monotonic Neural Networks

**Aishwarya Sivaraman**
University of California, Los Angeles
dcssiva@cs.ucla.edu

**Golnoosh Farnadi**
Mila/Université de Montréal
farnadig@mila.quebec

**Todd Millstein**
University of California, Los Angeles
todd@cs.ucla.edu

**Guy Van den Broeck**
University of California, Los Angeles
guyvdb@cs.ucla.edu

## Abstract

The widespread adoption of deep learning is often attributed to its automatic feature construction with minimal inductive bias. However, in many real-world tasks, the learned function is intended to satisfy domain-specific constraints. We focus on monotonicity constraints, which are common and require that the function's output increases with increasing values of specific input features. We develop a counterexample-guided technique to provably enforce monotonicity constraints at prediction time. Additionally, we propose a technique to use monotonicity as an inductive bias for deep learning. It works by iteratively incorporating monotonicity counterexamples in the learning process. Contrary to prior work in monotonic learning, we target general ReLU neural networks and do not further restrict the hypothesis space. We have implemented these techniques in a tool called COMET.[1] Experiments on real-world datasets demonstrate that our approach achieves state-of-the-art results compared to existing monotonic learners, and can improve the model quality compared to those that were trained without taking monotonicity constraints into account.

## 1 Introduction

Deep neural networks are increasingly used to make sensitive decisions, including financial decisions such as whether to give a loan to an applicant [25] and as controllers for safety critical systems such as autonomous vehicles [7, 54]. In these settings, for safety, ethical, and legal reasons, it is of utmost importance that some of the decisions made are monotonic. For example, one would expect an individual with a higher salary to have a higher loan amount approved, all else being equal, and the speed of a drone to decrease with its proximity to the ground. Learning problems in medicine, revenue-maximizing auctions [17], bankruptcy prediction, credit rating, house pricing, etc., all have monotonicity as a natural property to which a model should adhere. Guaranteeing monotonicity helps users to better trust and understand the learned model [24]. Furthermore, prior knowledge about monotonic relationships can also be an effective regularizer to avoid overfitting [14].

Unfortunately, there is no easy way to specify that a trained neural network should be monotonic in one or more of its features. Existing approaches to this problem, such as min-max networks [40], monotonic lattices [16], and deep lattice networks [53], guarantee monotonicity by construction but do so at the cost of significantly restricting the hypothesis class. Other solutions, such as learning a linear function with positive coefficients, are even more restrictive. Furthermore, techniques that

enforce monotonicity as a soft constraint in neural networks [41, 23] suffer from not being able to provide any provable monotonicity guarantee at prediction time. Finally, the well-known framework of isotonic regression [4, 37] is effective only when the training data can be partially ordered, which is rarely the case in high dimensions.

This paper develops techniques to incorporate monotonicity constraints for standard ReLU neural networks without imposing further restrictions on the hypothesis space. These techniques leverage recent work that employs automated theorem provers to formally verify robustness and safety properties of neural networks [49, 50, 18, 28]. First, we present a counterexample-guided algorithm that provably guarantees monotonicity at prediction time, given an arbitrary ReLU neural network. Our approach works by constructing a *monotonic envelope* of the given model on-the-fly via verification counterexamples. Empirically we show that we guarantee monotonicity with little to no loss in model quality at a computational cost on the order of a few seconds on standard datasets. Second, we propose a new counterexample-guided algorithm to incorporate monotonicity as an inductive bias during training. We identify monotonicity counterexamples on the training data, inducing additional supervision for training the network, and perform this process iteratively. We also show that monotonicity is an effective regularizer: our counterexample-guided learning algorithm improves the overall model quality. Empirically, the two algorithms, when used in conjunction, enable better generalization while guaranteeing monotonicity for both regression and classification tasks. We have implemented our algorithms in a tool called "COunterexample-guided Monotonicity Enforced Training" (COMET). Finally, we demonstrate that COMET outperforms min-max and deep lattice networks [53] on four real-world benchmarks.

**Organization.** Section 2 introduces our problem statement and notation. Sections 3 and 4 respectively describe our proposed algorithms: counterexample-guided monotonic prediction and counterexample-guided monotonic training. Experimental results in Sections 3.2 and 4.2 demonstrate the potential of COMET on real-world benchmark datasets. Section 5 reviews related work in learning monotonic functions. We conclude and provide future directions in Section 6.

## 2 Preliminaries: Finding Monotonicity Counterexamples

We begin by introducing some common notation. Let $\mathcal{X}$ be the input space consisting of $d$ features, and suppose that it is a compact finite subset $\mathcal{X} = [L, U]^d$ of $\mathbb{R}^d$. Let $\mathcal{Y}$ be the output space. We consider regression and (probabilistic) binary classification tasks where $\mathcal{Y}$ is totally ordered.

Our goal will be to learn functions that are monotonic in some of their input features.

**Definition 1.** A function $f : \mathcal{X} \to \mathcal{Y}$ is *monotonically increasing* in features $S$ iff each feature in $S$ is totally ordered and for any two inputs $x, x' \in \mathcal{X}$ that are (i) non-decreasing in features $S$, $\forall i \in S, x[i] \leq x'[i]$, and (ii) holding all else equal, $\forall k \notin S, x[k] = x'[k]$, the output of the function is non-decreasing: $f(x) \leq f(x')$.

Formal properties of functions are often characterized in terms of their counterexamples. Counterexample-guided algorithms are prevalent in the field of formal methods, for example to verify [10] and synthesize programs [44]. The techniques proposed in this paper will be centered around using counterexamples to the monotonicity specification.

**Definition 2.** A pair of inputs $x, x' \in \mathcal{X}$ is a *monotonicity counterexample pair* for the $i$th feature of function $f : \mathcal{X} \to \mathcal{Y}$ iff the points are (i) non-decreasing in feature $i$, that is, $x[i] \leq x'[i]$, (ii) holding all else equal, that is, $\forall k \neq i, x[k] = x'[k]$, and (iii) the function is decreasing: $f(x) > f(x')$.

Notably, for a function to be (jointly) monotonic in features $S$, it is both necessary and sufficient that there does not exist a monotonicity counterexample pair for any of the individual features in $S$.

ReLU neural networks generalize well and are widely used [20, 51, 13], particularly in the context of verification and robustness. Hence, we will assume that $f$ is a ReLU neural network.

**Definition 3.** A *ReLU neural network* is a directed acyclic computation graph consisting of neurons that compute $\text{ReLU}(\sum_i w_i x_i + b)$, where the activation function is a rectified linear unit $\text{ReLU}(y) = \max(0, y)$, the weights $w_i$ and bias $b$ are parameters associated with each neuron, and neuron inputs $x_i$ are either input features or values of other neurons. The value of a designated output neuron defines the value of a function $f : \mathcal{X} \to \mathcal{Y}$.

Counterexample-guided algorithms rely on the ability to *find* counterexamples, usually by relegating the task to an off-the-shelf solver. This requires that both the counterexample specification and the object of interest — in this case the function $f$ — can be encoded in a formal language amenable to automated reasoning. We will use a *satisfiability modulo theories* (SMT) solver [5] for this purpose. Recall that satisfiability (SAT) is the problem of deciding the existence of assignments of truth values to variables such that a propositional logical formula is satisfied. SMT generalizes SAT to deciding satisfiability for formulas with respect to a decidable background theory [5]. We will use the background theory of linear real arithmetic (LRA), which allows for expressing Boolean combinations of linear inequalities between real number variables.

The encoding of ReLU neural networks into SMT(LRA) is well-known and readily available [28, 27]. Briefly, the relationship between any neuron value and its inputs is encoded in SMT(LRA) as follows. The linear sum over neuron inputs is already a linear constraint. Additionally, we encode the non-linearity of the ReLU activation function using logical implications in SMT. Concretely, for $z = \mathrm{ReLU}(y) = \max(0, y)$, we add two SMT constraints: $y > 0 \rightarrow z = y$ and $y \leq 0 \rightarrow z = 0$.

We can now ask an SMT solver to find monotonicity counterexample pairs: we simply take the (linear) conditions in Definition 2 and conjoin with the SMT (LRA) encoding of the function $f$. Linear real arithmetic is a decidable theory [45]; hence we will always obtain a correct counterexample if one exists. In Section 3, we require the ability to obtain a counterexample that maximally violates the monotonicity specification. Hence, we use Optimization Modulo Theories (OMT) [38], which is an extension of SMT for finding models that optimize secondary linear objectives, which is again decidable. Note that our definitions consider monotonically increasing features, and we assume that form of monotonicity throughout. We can analogously define corresponding notions for monotonically decreasing features, and our algorithms can be applied straightforwardly to that setting as well.

While this setup allows us to verify monotonicity of a learned function, it is not at all clear how to *guarantee* monotonicity, or how to enforce monotonicity during training as an inductive bias. The next two sections present the counterexample-guided algorithms that address these challenges.

## 3    Counterexample-Guided Monotonic Prediction

A neural network trained using traditional approaches is not guaranteed to satisfy monotonicity constraints. In this section, we describe a technique to convert a non-monotonic model to a monotonic one. The technique leverages monotonicity counterexamples to construct a monotonic *envelope* (or *hull*) of the learned model. Further, our technique is *online*: the monotonic envelope is constructed on-the-fly at prediction time.

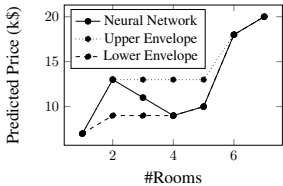

Figure 1: Monotone envelopes around a simple non-monotone learned function

As an example, consider the regression task of predicting house prices, which monotonically increase with the number of rooms. Suppose that the solid line (——) in Figure 1 plots the learned model's predictions. This function is not monotonic; for example $f(3) > f(4)$. The two dotted lines in Figure 1 show two monotonic envelopes that our technique produces: an *upper* envelope ($\cdots$) that increases the output where necessary to ensure monotonicity, and a *lower* envelope (- - -) that decreases the output where necessary to ensure monotonicity. The rest of this section describes these envelopes formally and presents an empirical evaluation of the technique.

### 3.1    Envelope Construction

We first describe envelope construction for the case with a single monotonic feature (with any number of other features) and then generalize the approach to handle multiple monotonic features.

#### 3.1.1    Envelope - Single Monotonic Feature

Recall that Definition 2 in the previous section defines when a pair of inputs constitutes a monotonicity counterexample. To construct the envelope we require a special form of such counterexamples, namely *maximal* ones in terms of the degree of monotonicity violation, while fixing a single input example.

**Definition 4.** Consider example $x \in \mathcal{X}$, function $f : \mathcal{X} \to \mathcal{Y}$, and feature $i$. Let set $\mathcal{A}$ (resp. $\mathcal{B}$) consist of all examples $x'$ such that $(x, x')$ (resp. $(x', x)$) is a counterexample pair for $f$ and $i$. Then, a *lower envelope counterexample* for example $x$, function $f$ and feature $i$ is an example $x' \in \mathcal{A}$ that minimizes $f(x')$. An *upper envelope counterexample* is an example $x' \in \mathcal{B}$ that maximizes $f(x')$.

For example, consider Figure 1 again. The upper envelope counterexample for input 3 is the input 2, since $f(2)$ has the maximal value of all counterexamples below 3. The lower envelope counterexample for the input 3 is 4, since $f(4)$ has the minimal value of all counterexamples above 3.

Now we can define the upper and lower envelopes of a function.

**Definition 5.** The *upper envelope* $f_i^u$ of function $f : \mathcal{X} \to \mathcal{Y}$ for feature $i$ is defined as follows:

$$f_i^u(x) = \begin{cases} f(x') & \text{where } x' \text{ is an upper envelope counterexample for } x, f, \text{ and } i \\ f(x) & \text{if no such counterexample exists} \end{cases}$$

The *lower envelope* $f_i^l$ is defined analogously.

We observe that it is not necessary to construct the envelope function explicitly. Rather, to ensure monotonicity, it suffices to construct the envelope incrementally at prediction time. Given an input $x^t$, we make a single query to an SMT solver to find the input's upper (lower) envelope counterexample or determine that no such counterexample exists. Note that this query is much simpler than would be required to verify that the original function is monotonic. Doing the latter would require searching for an arbitrary monotonicity counterexample pair (Definition 2), which is a pair of points. In contrast, our query is given the input $x^t$ and hence only requires the SMT solver to search over the space of inputs that are identical to $x^t$ except in the $i$th dimension. Concretely, for a feature $i$ in the bounded interval $[L, U]$, the upper envelope search is over the interval $[L, x^t[i])$ and the lower envelope search is over the interval $(x^t[i], U]$. Empirically we will later show that our envelope construction is faster than querying for an arbitrary counterexample pair (see Figure 3).

### 3.1.2 Envelope - Multi-Dimensional Case

We now generalize our envelope construction to the case where multiple dimensions are monotonic. For space reasons we present only the upper envelope construction; the lower envelope is analogous.

Recall from Section 2 that, to verify if a function is monotonic in more than one dimension, it is sufficient to verify that it is monotonic in each dimension separately. However, to construct the envelope, it is not sufficient to identify maximal counterexamples in each dimension and then take the maximum of these maxima. The envelopes produced using that approach are not guaranteed to be monotonic (see appendix for an example). To overcome this problem, we generalize to multiple dimensions by searching *jointly* in all monotonic dimensions and prove that this approach is correct.

**Definition 6.** Consider example $x \in \mathcal{X}$, function $f : \mathcal{X} \to \mathcal{Y}$, and set of features $S$. Let set $\mathcal{B}$ consist of all examples $x'$ such that $\forall i \in S, x'[i] \leq x[i]$ and $\forall i \notin S, x'[k] = x[k]$ and $f(x') > f(x)$. An *upper envelope counterexample* is an example $x' \in \mathcal{B}$ that maximizes $f(x')$.

It is easy to show that this approach does not identify spurious counterexamples: if an upper envelope counterexample exists for $x$ and $f$ and set of features $S$, then there is a dimension $i \in S$ and points $x'$ and $x''$ such that $x'$ and $x''$ are a monotonicity counterexample for $f$ in the $i$th dimension.

We can now define the upper envelope function, analogous to the single-dimensional case:

**Definition 7.** The *upper envelope* $f_S^u$ of function $f : \mathcal{X} \to \mathcal{Y}$ for feature set $S$ is defined as follows:

$$f_S^u(x) = \begin{cases} f(x') & \text{where } x' \text{ is an upper envelope counterexample for } x, f, \text{ and } S \\ f(x) & \text{if no such counterexample exists} \end{cases}$$

Finally, we prove that the upper envelope is in fact monotonic, even when the function $f$ is not.

**Theorem 1.** For any function $f$ and set of features $S$, the *upper envelope* $f_S^u$ is monotonic in $S$.

*Proof.* Let $i_0 \in S$ and $x$ and $x'$ be any two inputs such that $x[i_0] \leq x'[i_0]$ and $\forall k \neq i_0, x[k] = x'[k]$. We will prove that $f_S^u(x) \leq f_S^u(x')$ and hence that $f_S^u$ is monotonic. There are two cases:

1. An input $x'_e$ is the upper envelope counterexample for $x'$, $f$, and $S$, so $f^u_S(x') = f(x'_e)$. We have two subcases.

   - An input $x_e$ is the upper envelope counterexample for $x$, $f$, and $S$, so $f^u_S(x) = f(x_e)$. By Definition 6 we have that $\forall i \in S, x_e[i] \leq x[i] \wedge \forall i \notin S, x_e[k] = x[k]$, so also $\forall i \in S, x_e[i] \leq x'[i] \wedge \forall i \notin S, x_e[k] = x'[k]$. Therefore again by Definition 6 it must be the case that $f(x_e) \leq f(x'_e)$.
   - There is no upper envelope counterexample for $x$, $f$, and $S$, so $f^u_S(x) = f(x)$. Since $\forall i \in S, x[i] \leq x'[i] \wedge \forall i \notin S, x[k] = x'[k]$, by Definition 6 it must be the case that $f(x) \leq f(x'_e)$.

2. There is no upper envelope counterexample for $x'$, $f$, and $S$. The proof is similar (details in appendix). $\square$

Hence, our envelope construction algorithm guarantees monotonicity of the predictive function, regardless of where it is evaluated, and regardless of the underlying learned function.

### 3.2 Empirical Evaluation of Monotonic Envelopes

We report the experimental results on the quality and performance of the envelope construction algorithm. Experiments were implemented in Python using the Keras deep learning library [9], we use the ADAM optimizer [29] to perform stochastic optimization of the neural network models, and we use the Optimathsat [39] solver for counterexample generation.

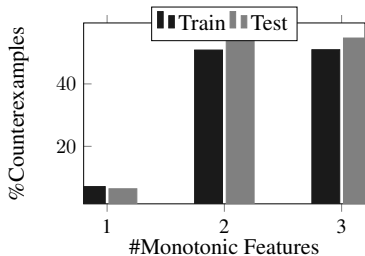

Figure 2: Empirically, the best learned baseline model is not monotonic. The figure presents the percentage of examples that have an upper or lower envelope counterexample for the *Auto MPG* dataset.

**Data and experiment setup**: We use four datasets: *Auto MPG* and *Boston Housing* are regression datasets used for predicting miles per gallon (monotonically decreasing with respect to features *weight (W)*, *displacement (D)*, and *horsepower (HP)*) and housing prices (monotonically decreasing in *crime rate* and increasing in *number of rooms*) respectively and are obtained from the UCI machine learning repository [6]; *Heart Disease* [19] and *Adult* [6] are classification datasets used for predicting the presence of heart disease (monotonically increasing with *trestbps (T)*, *cholestrol (C)*) and income level (monotonically increasing with *capital-gain* and *hours per week*) respectively. For each dataset, we identify the best baseline architecture and parameters by conducting grid search and learn the best ReLU neural network (NN$_b$). We carry out our experiments on three random 80/20 splits and report average test results, except for the Adult dataset, for which we report on one random split.

**Q1. Is a deep neural network trained on such data monotonic?** Figure 2 shows that the best baseline model (NN$_b$) is not monotonic, motivating the need for *envelope* predictions that guarantee monotonicity. The percentage of data points that have a counterexample can be as high as 50% for *Auto MPG*. See Table 6 in the appendix for detailed results on all datasets, where the percentage can be as high as 98%.

Table 1: For regression (MSE, Left Table) and classification (Accuracy, Right Table) datasets, envelope predictions on test data have similar quality compared to baseline models. This means we can guarantee monotonic predictions with little to no loss in model quality.

| Dataset | Feature | NN$_b$ | Envelope |
|---|---|---|---|
| Auto-MPG | Weight | 9.33±3.22 | **9.19±3.41** |
| | Displ. | 9.33±3.22 | 9.63±2.61 |
| | W,D | 9.33±3.22 | 9.63±2.61 |
| | W,D,HP | 9.33±3.22 | 9.63±2.61 |
| Boston | Rooms | 14.37±2.4 | **14.19±2.28** |
| | Crime | 14.37±2.4 | **14.02±2.17** |

| Dataset | Feature | NN$_b$ | Envelope |
|---|---|---|---|
| Heart | Trestbps | 0.85±0.04 | 0.85±0.04 |
| | Chol. | 0.85±0.04 | 0.85±0.05 |
| | T,C | 0.85±0.04 | 0.85±0.05 |
| Adult | Cap. Gain | 0.84 | 0.84 |
| | Hours | 0.84 | 0.84 |

**Q2. When enforcing monotonicity using an envelope, does it come at a cost in terms of prediction quality?** In this experiment, we compare the quality of the original model (NN$_b$) with its

envelope on the test data. We select the envelope with the lowest train mean squared error (MSE) in case of regression and highest train accuracy in case of classification. Table 1 demonstrates that an envelope can be used with a single or multiple monotonic features with little to no loss in prediction quality. In fact, in some cases (see rows in **bold**), the envelope has better average quality. This can be explained as follows: although the true data distribution is naturally monotonic, existing learning algorithms might be missing simpler monotonic models and instead overfit a non-monotonic function because of noise in the training data.

**Q3. How scalable is on-the-fly envelope construction?** In this experiment, we report the run times for the *Auto MPG* dataset. Recall that the envelope approach need only search for maximal counterexamples relative to a given input. Owing to the narrowed search space, we see that envelope prediction time is comparable to the baseline model's prediction time in smaller models

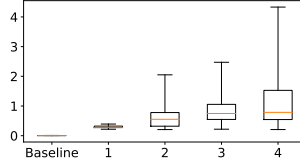
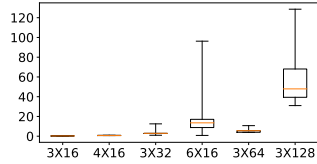

Figure 3: Prediction Time (s) vs. #Monotonic Features

Figure 4: Prediction Time (s) vs. Model Size

(see Figure 4). Overhead caused by envelope construction is only a *few seconds*. In contrast, the overhead to finding a maximal counterexample *pair* (Definition 2) for a single monotonic feature is 48.29 minutes. As a scalability study, in Figure 4, we plot the time taken to obtain a monotonic prediction for various model sizes. We can see that the envelope prediction time is comparable to the baseline prediction time in smaller models but grows with the model size. The growth is significantly less pronounced in the number of monotonic features (see Figure 3). Of course, when violating monotonicity leads to safety, ethical or legal problems, the question is not whether we can scale monotonicity enforcement, but whether it is safe to use machine learning at all. In this context, the computational price of enforcing monotonicity, even if it ends up being significant, is entirely warranted.

## 4   Counterexample-Guided Monotonicity Enforced Training

In this section we propose an algorithm that uses monotonicity as an inductive bias during learning to improve model quality. This algorithm is orthogonal to the envelope prediction technique of the previous section; we evaluate the learning algorithm both on its own and in conjunction with the envelope technique.

### 4.1   Counterexample-guided Learning

The learning algorithm consists of two phases that alternate: the training phase and the verification phase. The training phase is given labeled input data and produces the best candidate model $f$. The verification phase checks if a given model is monotonic; if not, it generates one or more counterexamples, which are provided as additional data for the next iteration of the training phase. These two phases repeat for $T$ epochs, which is a hyperparameter to the algorithm.

The algorithm is universal in the sense that it is compatible with any training technique that produces ReLU models and does not further restrict the hypothesis class. This gives our approach an advantage over prior monotonic learners [40, 53].

The verification phase could use Definition 2 to identify monotonicity counterexamples, but this has two major drawbacks: (1) it is computationally expensive as the size of the pre-trained model grows; (2) an arbitrary counterexample might include out-of-distribution examples, which are therefore not representative. Hence, we instead appeal to Definition 6 to generate maximal counterexamples relative to each training point. In each epoch, for each train point we generate and use both *upper* and *lower envelope counterexamples* as additional data for the next round of training.

At this point, we are almost done with the algorithm, with the following detail to address. Counterexamples generated by the verification procedure do not have a known ground-truth label. There are different heuristics that one could adopt to label these points and encourage the learned function to become more monotonic. In our algorithm, for regression tasks we calculate the average prediction

values of upper and lower counterexamples and the given training point and assign this average as the label for these counterexamples and the training point. The hypothesis is that using the average value will result in a smoother loss with respect to monotonicity. For classification tasks, we assign each counterexample point the same label as the corresponding training point. Empirically (see Table 3), we will show that this labelling heuristic is sufficient to improve the model quality.

Data augmentation through counterexamples could cause drift in the model quality. Our approach guards against this in multiple ways. First, data augmentation with counterexamples is recomputed for each batch at every epoch. This ensures that: 1) an incorrect old counterexample does not burden the learning, and 2) learning incorporates multiple counterexamples at a time and so is less sensitive to any particular one. Second, the labeling heuristic for counterexamples provides a smoother loss with respect to monotonicity. Empirically (see Table 2), we will show that there is no drift in the model quality. The quality of our model is similar or better than a model trained without monotonicity constraints.

## 4.2 Empirical Evaluation of COMET

We will now evaluate our iterative algorithm for training with monotonicity counterexamples, as well as the entire COMET pipeline, which also includes the envelope technique from the previous section. We use the same datasets as in Section 3.2.

Table 2: Monotonicity is an effective inductive bias. Counterexample-guided Learning (CGL) improves the quality of the baseline model in regression (MSE, Left Table) and classification (Accuracy, Right Table) datasets

| Dataset | Feature | $NN_b$ | CGL |
|---|---|---|---|
| Auto-MPG | Weight | 9.33±3.22 | **9.04±2.76** |
| | Displ. | 9.33±3.22 | **9.08±2.87** |
| | W,D | 9.33±3.22 | **8.86±2.67** |
| | W,D,HP | 9.33±3.22 | **8.63±2.21** |
| Boston | Rooms | 14.37±2.4 | **12.24±2.87** |
| | Crime | 14.37±2.4 | **11.66±2.89** |

| Dataset | Feature | $NN_b$ | CGL |
|---|---|---|---|
| Heart | Trestbps | 0.85±0.04 | **0.86±0.02** |
| | Chol. | **0.85±0.04** | 0.85±0.05 |
| | T,C | 0.85±0.04 | **0.86±0.06** |
| Adult | Cap. Gain | **0.84** | **0.84** |
| | Hours | **0.84** | **0.84** |

**Q4. Is the stronger inductive bias of our learning algorithm able to improve the overall quality of the original non-monotonic model?** In this experiment we compare the test quality of the model learned with monotonicity counterexamples with the original model ($NN_b$). From Table 2, we can see that monotonicity is indeed an effective inductive bias that helps improve the model quality. It is able to reduce the error on all regression datasets, with the biggest decrease from 14.37 to 11.66 for the *Boston Housing* dataset when employing monotonicity counterexamples based on the Crime Rate feature. Although the algorithm improves the quality, it does not guarantee monotonic predictions.

**Q5. Does our learning algorithm make the original non-monotonic model more monotonic?** To quantify if a function is more monotonic, we calculate the reduction in the number of counterexamples. On average, our algorithm reduces the number of test counterexamples by 62%. Although in some cases we can remove all counterexamples, in general this is not the case (see Table 7 in Appendix for detailed results). This motivates the need for using monotonic *envelopes* (described in Section 3) in conjunction with the counterexample-guided learning algorithm, to guarantee monotonic predictions.

Table 3: For regression (MSE, Left Table) and classification (Accuracy, Right Table) datasets, counterexample-guided learning improves the envelope quality

| Dataset | Features | $NN_b$ Env. | COMET |
|---|---|---|---|
| Auto-MPG | Weight | 9.19±3.41 | **8.92±2.93** |
| | Displ. | 9.63±2.61 | **9.11±2.25** |
| | W,D | 9.63±2.61 | **8.89±2.29** |
| | W,D,HP | 9.33±2.61 | **8.81±1.81** |
| Boston | Rooms | 14.19±2.28 | **11.54±2.55** |
| | Crime | 14.02±2.17 | **11.07±2.99** |

| Dataset | Features | $NN_b$ Env. | COMET |
|---|---|---|---|
| Heart | Trestbps | 0.85±0.04 | **0.86±0.03** |
| | Chol. | 0.85±0.05 | **0.87±0.03** |
| | T,C | 0.85±0.05 | **0.86±0.03** |
| Adult | Cap. Gain | **0.84** | **0.84** |
| | Hours | **0.84** | **0.84** |

**Q6. Does counterexample-guided learning help improve the quality of the original model's envelope?** In Section 3.2 Q2, (Table 1), we showed that the envelope has similar model quality compared to the baseline model. By additionally enforcing monotonicity constraints through counterexample-guided re-training, we further improve the envelope quality (Table 3). In this experiment we re-train $NN_b$ with counterexamples for 40 epochs, model selection is based on train

quality, and we report the change in the quality of the test envelope (see the appendix for additional model selection experiments). Thus, we get both a monotonicity guarantee and better generalization performance.

Table 4: COMET outperforms Min-Max networks on all datasets. COMET outperforms DLN in regression datasets and achieves similar results in classification datasets.

| Dataset | Features | Min-Max | DLN | COMET |
|---|---|---|---|---|
| Auto-MPG | Weight | 9.91±1.20 | 16.77±2.57 | **8.92±2.93** |
| | Displ. | 11.78±2.20 | 16.67±2.25 | **9.11±2.25** |
| | W,D | 11.60±0.54 | 16.56±2.27 | **8.89±2.29** |
| | W,D,HP | 10.14±1.54 | 13.34±2.42 | **8.81±1.81** |
| Boston | Rooms | 30.88±13.78 | 15.93±1.40 | **11.54±2.55** |
| | Crime | 25.89±2.47 | 12.06±1.44 | **11.07±2.99** |

| Dataset | Features | Min-Max | DLN | COMET |
|---|---|---|---|---|
| Heart | Trestbps | 0.75±0.04 | 0.85±0.02 | **0.86±0.03** |
| | Chol. | 0.75±0.04 | 0.85±0.04 | **0.87±0.03** |
| | T,C | 0.75±0.04 | **0.86±0.02** | **0.86±0.03** |
| Adult | Cap. Gain | 0.77 | **0.84** | **0.84** |
| | Hours | 0.73 | **0.85** | 0.84 |

**Q7. How does the performance of** COMET **compare to existing work?** Table 4 reports the MSE and accuracy of COMET compared to two existing methods that guarantee monotonicity: min-max networks [12] and deep lattice networks (DLN) [53]. We tune Adam stepsize, learning rate, number of epochs, and batch size on all methods. Additionally, for DLN we tune calibration keypoints and report the results based on the six-layer architecture as proposed by the authors. The results in Table 4 indicate that COMET outperforms min-max networks on all datasets and DLN on all except for Adult, where we are similar.

**Q8. How robust is** COMET **to data outliers?** COMET constructs its monotonic envelope on the learned function and not on the data. Therefore, individual data outliers will not affect it too much. Moreover, if the function to be learned is naturally monotonic, enforcing invariants counteracts noise and outliers, leading to improved robustness. To illustrate this advantage, we duplicate 1% of the data and modify the value of the monotonic feature and the label for each new point in order to introduce monotonicity outliers (violations). For example, for an increasing monotonic feature, we reduce the label and increase the value of the monotonic feature. Table 5 shows that our approach produces *more robust models*, with COMET improving baseline model quality.

Table 5: With monotonicity data outliers, COMET produces models that are more robust than the baseline models ($NN_b$) for regression (MSE, Left Table) and classification (Accuracy, Right Table) datasets.

| Dataset | Features | $NN_b$ | COMET |
|---|---|---|---|
| Auto-MPG | Weight | 13.54±4.65 | **10.50±1.87** |
| | Displ. | 12.00±2.94 | **10.34±1.25** |
| | W,D | 15.35±2.30 | **13.84±3.09** |
| | W,D,HP | 10.26±2.19 | **9.48±1.29** |
| Boston | Rooms | 12.79±3.88 | **10.23±1.95** |
| | Crime | 21.13±4.41 | **19.20±6.64** |

| Dataset | Features | $NN_b$ | COMET |
|---|---|---|---|
| Heart | Trestbps | 0.77±0.07 | **0.78±0.07** |
| | Chol. | **0.77±0.06** | **0.77±0.06** |
| | T,C | 0.77±0.06 | **0.81±0.03** |
| Adult | Cap. Gain | **0.82** | **0.82** |
| | Hours | **0.82** | **0.82** |

## 5   Related Work

**Monotonic Networks.** Liu et al. [31] propose a concurrent work that uses verification techniques to learn certified monotonic networks. The approach encodes an arbitrary ReLU neural network using mixed-integer linear programming and solves an optimization problem to verify monotonicity. The optimization problem is to identify if the minimum derivative of the function is non-negative. Further, the approach learns monotonic networks by training with heuristic monotonicity regularizations and gradually increasing the regularization magnitude until it passes the monotonicity verification. We differ from this work in two ways. First, our envelope technique produces a monotonic version of an arbitrary ReLU neural network without having to retrain it. Second, we solve an optimization problem to identify the maximum violation for a given point, which is necessary for the envelope construction.

Other related work in this area can be categorized into algorithms that (1) guarantee monotonicity by restricting the hypothesis space, or (2) incorporate monotonicity during learning without any guarantees. In the first category, Archer and Wang [3] propose a monotone model by constraining the neural net weights to be positive. Other methods enforce constraints on model weights [11, 46, 33, 15, 2] and force the derivative of the output to be strictly positive [47]. Monotonic networks [40]

guarantee monotonicity by constructing a three-layer network using monotonic linear embedding and max-min-pooling. Daniels and Velikova [12] generalized this approach to handle functions that are partially monotonic. Deep lattice networks (DLN) [53] proposed a combination of linear calibrators and lattices for learning monotonic functions. Lattices are structurally rigid thereby restricting the hypothesis space significantly. Our envelope technique is similar to these works in that it guarantees monotonicity. However, it does so at prediction time and can do so for any ReLU neural networks, without needing to restrict the hypothesis space further. Finally, isotonic regression [4, 37] requires the training data to be partially ordered, which is unlikely to happen; in general input points over many features are not partially ordered.

In the second category, monotonicity can be incorporated in the learning process by modifying the loss function or by adding additional data. Monotonicity Hints [41] proposes a modified loss function that penalizes non-monotonicity of the model. The algorithm models the input distribution as a joint Gaussian estimated from the training data and samples random pairs of monotonic points that are added to the training data. Gupta et. al. [23] introduce a point-wise loss function that acts as a soft monotonicity constraint. Our approach is similar to these works in that it adds additional data to enforce monotonicity. However, COMET's counterexample-guided learning and envelope technique together guarantee monotonicity, while these works provide no such guarantees. In addition, unlike prior work, we look beyond the neighborhood of a training point by identifying maximal violations. Other works enforce monotonicity to accelerate learning of probabilistic models in data-scarce and knowledge-rich domains [34, 1, 52]. Similarly, these works fail to provide any formal guarantee for the learned model.

**Verification of Neural Networks and Adversarial Learning.** Reluplex [28], an augmented SMT solver, verifies properties of networks with ReLU activation functions. Huang et. al. [27] leverage SMT for verification of safety properties by discretizing the continuous region around an input and show that there are no counterexamples. Our approach leverages the SMT encodings of neural networks from this prior work but uses them only to obtain counterexamples rather than for verification. Recently, many approaches propose adversarially robust algorithms which can be divided into empirical [30, 32, 21, 22] and certified defenses [48, 43, 35, 26, 36, 42, 18]. We are closely related to these works, in that we carry out adversarial training using counterexamples. However, we differ in two ways. First, to the best of our knowledge, there is no related work in the adversarial robustness literature for ensuring monotonicity. Second, related work in adversarial training only ensures correctness in the neighborhood of a training point, while we globally search for a counterexample and are able to discover long-range monotonicity violations. Counterexample-driven learning has also been used to enforce fairness constraints on Bayesian classifiers [8].

# 6 Conclusions & Future Work

We presented two algorithms that incorporate monotonicity constraints into neural networks: counterexample-guided prediction that guarantees monotonicity and counterexample-guided training that enforces monotonicity as an inductive bias. We demonstrate the effectiveness of these techniques on regression and classification tasks.

In the future, we plan to further increase the scalability of COMET and study how to modify neural network learning algorithms in order to enable that scalability. Moreover, scalability could potentially be improved by using specialized SMT solvers, or by using MaxSMT solvers instead of OMT for finding maximal counterexamples. Our approach is not limited to ReLU activation functions: other activation functions with an SMT encoding are amenable to counterexample-guided learning. Another interesting direction for future work is to study other types of inductive bias, such as those coming from algorithmic fairness. We plan to explore new strategies that use counterexamples to learn with and enforce these desirable properties.

**Acknowledgments**   The authors would like to thank Murali Ramanujam, Sai Ganesh Nagarajan, Antonio Vergari, and Ashutosh Kumar for helpful feedback on this research.

**Broader Impact**   In scenarios where monotonicity is a natural and fair requirement on the learned function, it is clear that eliminating errors or noise coming from monotonicity violations can increase the fairness of the predictions, and eliminate some frustration with receiving arbitrary outcomes. At the same time, one can imagine someone requiring monotonicity of certain features that should not

be monotonic, and the work in this paper can enable such undesirable behavior. By incorporating correct and ethical domain knowledge using our algorithms, we make the learned model more robust. We acknowledge that the counterexamples generated based on data can be used in ways other than the ways mentioned in this paper. Although our approach can produce monotonic predictions, it is still based on a model produced by a machine learning algorithm. Therefore, it becomes essential to understand that the monotonic model could also suffer from the same disadvantages as the original model, and reinforce the same biases. Hence, the user must be aware of such a system's limitations, especially when using these models to replace people in decision making.

**Funding Disclosure**   This work is supported in part by NSF grants CCF-1837129, IIS-1943641, IIS-1633857, DARPA XAI grant #N66001-17-2-4032, Sloan and UCLA Samueli Fellowships, and gifts from Intel and Facebook Research. Golnoosh Farnadi is supported by a postdoctoral scholarship from IVADO through the Canada First Research Excellence Fund (CFREF) grant.

## Footnotes

[1] https://github.com/AishwaryaSivaraman/COMET

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
