[Supplementary Material]

# A Appendix

## A.1 Envelope - Multi-Dimensional Case

Recall from Section 3.1.2 that to construct an envelope for a function $f$ and $S$ set of features, it is not sufficient to identify maximal counterexamples in each dimension and then take the maximum of these maxima. The envelopes produced using this approach are not guaranteed to be monotonic, which we now demonstrate with an example. Consider a function $f$ that is intended to be monotonically increasing in its two input features. Now, consider the point $(3, 5)$, suppose that $(1, 5)$ and $(3, 3)$ are the upper envelope counterexamples in each dimension (Definition 4), and suppose that $f(3, 3) > f(1, 5)$ so we set $f_{\{0,1\}}^u(3, 5) = f(3, 3)$. Now consider a second point $(7, 5)$, suppose that $(1, 5)$ and $(7, 2)$ are the upper envelope counterexamples in each dimension, and suppose that $f(1, 5) > f(7, 2)$ so we set $f_{\{0,1\}}^u(7, 5) = f(1, 5)$. Since $f(3, 3) > f(1, 5)$ we have that $f_{\{0,1\}}^u(3, 5) > f_{\{0,1\}}^u(7, 5)$, which violates monotonicity.

Therefore in the multi-dimensional case we search for counterexamples *jointly* over all monotonic dimensions (Definition 6). We now prove the correctness of this approach.

**Theorem 1.** For any function $f$ and set of features $S$, the *upper envelope* $f_S^u$ is monotonic in $S$.

*Proof.* Let $i_0 \in S$ and $x$ and $x'$ be any two inputs such that $x[i_0] \leq x'[i_0]$ and $\forall k \neq i_0, x[k] = x'[k]$. We will prove that $f_S^u(x) \leq f_S^u(x')$ and hence that $f_S^u$ is monotonic. There are two cases:

- An input $x_e'$ is the upper envelope counterexample for $x'$, $f$, and $S$, so $f_S^u(x') = f(x_e')$ and by Definition 6 $f(x') < f(x_e')$. We have two subcases.

  - An input $x_e$ is the upper envelope counterexample for $x$, $f$, and $S$, so $f_S^u(x) = f(x_e)$. By Definition 6 we have that $\forall i \in S, x_e[i] \leq x[i] \wedge \forall i \notin S, x_e[k] = x[k]$, so also $\forall i \in S, x_e[i] \leq x'[i] \wedge \forall i \notin S, x_e[k] = x'[k]$. Therefore again by Definition 6 either $f(x_e) \leq f(x')$ or $f(x_e) \leq f(x_e')$. Since $f(x') < f(x_e')$, either way we have that $f(x_e) \leq f(x_e')$.
  - There is no upper envelope counterexample for $x$, $f$, and $S$, so $f_S^u(x) = f(x)$. Since $\forall i \in S, x[i] \leq x'[i] \wedge \forall i \notin S, x[k] = x'[k]$, by Definition 6 either $f(x) \leq f(x')$ or $f(x) \leq f(x_e')$. Since $f(x') < f(x_e')$, either way we have that $f(x) \leq f(x_e')$.

- There is no upper envelope counterexample for $x'$, $f$, and $S$, so $f_S^u(x') = f(x')$. We have two subcases.

  - An input $x_e$ is the upper envelope counterexample for $x$, $f$, and $S$, so $f_S^u(x) = f(x_e)$. By Definition 6 we have that $\forall i \in S, x_e[i] \leq x[i] \wedge \forall i \notin S, x_e[k] = x[k]$, so also $\forall i \in S, x_e[i] \leq x'[i] \wedge \forall i \notin S, x_e[k] = x'[k]$. Therefore again by Definition 6 it must be the case that $f(x_e) \leq f(x')$, or else $x'$ would have an upper envelope counterexample.
  - There is no upper envelope counterexample for $x$, $f$, and $S$, so $f_S^u(x) = f(x)$. Then again by Definition 6 it must be the case that $f(x) \leq f(x')$, or else $x'$ would have an upper envelope counterexample. $\square$

## A.2 Empirical Evaluation

In this section we provide additional experiment setup details and results from Section 3.2 and Section 4.2.

**System Specifications and experiment Setup:** All experiments were run on an Intel(R) Xeon(R) Gold 5220 CPU @ 2.20GHz CPU with 512GB of DDR3 RAM running Ubuntu 18.04.3 LTS with kernel 5.3.0-28-generic. Experiments were implemented in Python using the Keras deep learning library [9]. We use the ADAM optimizer [29] to perform stochastic optimization of the neural network models, and the Optimathsat [38] solver for counterexample generation. For each dataset, we train five baseline architectures from a set of configurations and choose the best architecture based on train error. For *Boston Housing*, *Heart Diseases*, and *Adult* dataset, best baseline architecture includes three layers and 16 hidden neurons per layer. For *Auto MPG* dataset, best baseline architecture

includes three layers and 12 hidden neurons per layer (see Table 8 for best baseline neural network parameters).

**Min-Max and Deep Lattice Network setup:** Min-Max networks [12] proposes a fixed, feedforward three-layer (two hidden layer) architecture. The first layer computes different linear combinations of input that are partitioned into different groups. If increasing monotonicity is desired, then all weights connected to that input are constrained to be positive. Corresponding to each group, the second layer computes the maximum, and the final layer computes the minimum over all groups. For monotone features that are decreasing, we negate the feature to use the same architecture. The Deep Lattice Network [52] architecture consists of six layers as proposed by the authors: calibrators, linear embedding, calibrators, ensemble of lattices, calibrators, and linear embedding. Note that for these approaches, for each dataset, we tune parameters separately for each combination of monotonic features at each fold using grid search; hence it is optimized for each monotone prediction task. However, for COMET it is sufficient to tune parameters for the original neural network ($NN_b$) once per dataset.

Table 6: Here we present the results referred to in Q1. Empirically, the best baseline neural network model ($NN_b$) trained on data is not monotonic. The table presents the percentage of examples that have an upper or lower envelope counterexample.

| Dataset | Feature | Train | Test |
|---|---|---|---|
| | | % CG | % CG |
| Auto-MPG | Weight | 7.11 | 6.41 |
| | Displ. | 48.62 | 52.99 |
| | W,D | 50.85 | 54.7 |
| | W,D,HP | 50.96 | 54.7 |
| Boston Housing | Rooms | 7.59 | 7.92 |
| | Crime | 16.75 | 16.5 |
| Heart | Trestbps | 73.14 | 74.86 |
| | Chol. | 86.91 | 87.98 |
| | T,C | 97.38 | 98.91 |
| Adult | Cap. Gain | 1.57 | 1.39 |
| | Hours | 18.93 | 19.58 |

Table 7: Here we present the results referred to in Q5. Counterexample-guided learning (CGL) is able to make a model more monotonic by reducing the number of test and train counterexamples compared to the baseline model ($NN_b$). However, the algorithm is unable to guarantee monotonicity, motivating the need for monotonic *envelopes*.

| Dataset | Features | Train | | Test | |
|---|---|---|---|---|---|
| | | $NN_b$ | CGL | $NN_b$ | CGL |
| Auto-MPG | Weight | 22.33 | 11.33 | 5 | 2 |
| | Displ. | 139.67 | 37 | 37 | 10.33 |
| | W,D | 159.67 | 85.67 | 42.67 | 22.67 |
| | W,D,HP | 149.67 | 61.33 | 39.33 | 15 |
| Boston | Rooms | 30 | 15.67 | 8 | 6.33 |
| | Crime | 80 | 38.67 | 19 | 8 |
| Heart | Trestbps | 188.67 | 31 | 49 | 7 |
| | Chol. | 212.67 | 45.33 | 53 | 10.67 |
| | T,C | 235.67 | 169.67 | 60.33 | 40.33 |
| Adult | Cap. Gain | 7407 | 2755 | 1903 | 700 |
| | Hours | 379 | 0 | 84 | 0 |

Table 8: Best parameter configurations on each dataset for each data fold found using grid search for baseline neural networks ($NN_b$).

| | Auto-MPG | | | Boston | | | Heart | | | Adult | | |
|---|---|---|---|---|---|---|---|---|---|---|---|---|
| | Batch Size | # Epochs | LR | Batch Size | # Epochs | LR | Batch Size | # Epochs | LR | Batch Size | # Epochs | LR |
| 0 | 32 | 2000 | 0.01 | 64 | 1000 | 0.01 | 32 | 400 | 0.01 | 1024 | 500 | 0.01 |
| 1 | 32 | 1500 | 0.01 | 64 | 1000 | 0.001 | 32 | 400 | 0.01 | - | - | - |
| 2 | 32 | 2000 | 0.01 | 32 | 500 | 0.01 | 32 | 400 | 0.001 | - | - | - |

**Q6. Additional model selection experiment.**
In Section 4.2, model selection was based on minimum train error. In this experiment, we carry out model selection based on the least number of counterexamples. Overall, we find that monotonicity counterexamples act as a good inductive bias and improve model quality. However, there is a tradeoff on how much one could enforce monotonicity as a bias. Figure 5 plots test envelope MSE of *Auto MPG* and *Boston Housing* datasets. We can see that envelope construction on a function with minimum counterexamples has a higher error than the original model's envelope.

Figure 5: Monotonicity is a good inductive bias and helps in improving model accuracy. However, there is a tradeoff between performance and reducing the number of examples that have counterexamples.