[Reviews · NeurIPS 2020]

Review 1

Summary and Contributions: The paper discusses approach to have a prediction model which is monotonic with respect to certain input features. There are two orthogonal aspects discussed in a paper: 1) modifying the output of a neural network (limited to a dense ReLU network) for predictions, so that it is guaranteed to be monotonic 2) a data-augmentation technique during training, with the intention of making the model "more monotonic"

Strengths: The work is novel in the sense that it uses a different approach to enforce monotonicity, compared to prior approaches. The topic has been of some limited interest. They use SMT solvers at prediction time for making sure the output is monotonic. Instead of explicitly constructing a modified model, they compute the output for a specific input while still maintaining the monotonicity guarantee. They also show that using a similar approach to use as a data-augmentation technique for training, leads to better evaluation performance.

Weaknesses: Re Scalability: My concerns regarding scalability stay. Though the approach is novel, the models tried on are somewhat small (the largest model tried seems to be 3-layer/384 parameters, and the deepest one is 6-layer and only 96 parameters). Would be good if they add experiments to see how these scale up to be practical. It would be interesting to see how the performance is affected based on the number of layers, and total number of parameters. Figure 4 looks is in that direction, and a complete presentation of the results (prediction times as well as other metrics) across varying layers and parameters sizes, would make the work stronger.

Correctness: See weaknesses.

Clarity: The paper is well written.

Relation to Prior Work: Mostly, though I would like to see the authors to compare with prior with other approaches with respect to applicability. Does the work intend to claim that obsoletes prior approaches in all scenarios? If not, what are those? Stating the limitations more explicitly, would make this work stronger.

Reproducibility: Yes

Additional Feedback: I have read the author feedback.


Review 2

Summary and Contributions: This paper studies an interesting (and not very well-studied) problem: imposing monotonicity in learned models in ML (whether regression or classification). Monotonicity is an obvious constraint to impose on certain parameters (for example, income in credit rating). The authors study ReLU neural networks, but many of the ideas are not specific to this family of models.

Strengths: The good points of this paper is that it is well written and that it treats an interesting but fairly new problem. Furthermore, the combination of the 2 algorithms presented in the paper provides a formal guarantee of monotonicity. An interesting empirical finding is that imposing monotonicity can actually increase accuracy.

Weaknesses: However, the main disadvantage of the proposed approach is its computational complexity. Guaranteeing monotonicity requires solving an SMT instance at *prediction* time. This means not only that this technique is limited to small problems, but also that there is no guarantee that the algorithm will terminate in a reasonable time at prediction time (which sort of cancels out the formal guarantee of monotonicity). This algorithm also appears to lack robustness: one outlier (for example a high credit rating for someone with zero income) can affect the value of the model for all possible parameter values (in this example forcing everyone to have a high credit rating). This would be particularly dramatic in boolean classification where you could end up with a constant learnt function. Comment added after the rebuttal: the authors have satisfactorily replied to this point in their rebuttal. Of course, it is possible to just use the second algorithm on its own: this has a reasonable complexity but only encourages monotonicity rather than imposing it. I have some doubts about this approach since (if I have understood correctly) the training data is augmented with artificial examples in order to encourage monotonicity. This appears somewhat dangerous in that, after the addition of many such examples, there could conceivably be some systematic drift away from the original training data. Comment added after rebuttal: again the authros have satisfactorily replied to this point.

Correctness: The claims and the method are correct.

Clarity: The paper is well written (but see list of minor comments, below).

Relation to Prior Work: Yes

Reproducibility: Yes

Additional Feedback: Minor comments: line 21: all else being equal line 27: You say 'Unfortunately, there is no easy way to specify that a trained neural network should be monotonic' but it may be easy to impose a monotonicity constraint on the output function in formal-reasoning based methods. See, for example: Alexey Ignatiev, et al, A SAT-Based Approach to Learn Explainable Decision Sets. IJCAR 2018: 627-645. line 43: cost of the order of line 156: 'Recall..' This is not actually said explicitly in Section 2. Fig.2: I don't see what #Monotonic features means. In all the datasets you decsribe there are either 2 or 3 monotonic features, so where does the 1 case come from? line 221: 'The percentage of counterexamples...': What does this mean? That half of pairs of points are counterexamples or that in half the trials you found at least one counterexample or something else? Fig.4: What does model size mean? The total number of possible inputs or the number of parameters. line 232: finding a maximal line 246: then it generates Bilbiography: use {UCI}, {MLP}, {PXR}, {SMT}, Opti{M}ath{S}at, {ReLU} to impose capital letters; [11] and [12] is the same paper; ref [23] is incomplete and LLC Deepair is not an author; perhaps cite conference versions rather than arXiv versions (for example for [31],[34])


Review 3

Summary and Contributions: RE authors' feedback: I think this method is primarily useful for offline evaluation with no latency requirements. There are many such use cases justifying the relevance of this work. However, I think the authors should be clear and upfront about the scalability and latency issues with their proposed methods and mention that in the paper. The previous submission of this work that I reviewed did no have any timing details, and I'm happy the added that to the experimental section. My overall score remains the same. Note: I reviewed a previous version of this work. There are several updates and improvements to the paper since the last version. The paper presents a counter-example guided algorithm that guarantees monotonicity with respect to a feature at prediction time by constructing monotonic envelopes around the prediction function. It also presents a training algorithm that uses counterexamples to impose monotonicity during training as inductive bias. These counterexamples are generated using a solver on a formal description of the network operations and the monotonicity constraints. The paper is interesting to the ML community that deal with problems that include monotonicity constraints as domain knowledge or policy or ethical requirements. It introduces an algorithm that can be used on an already trained model with ReLU activations. The paper presents empirical results showing that the methods are effective on a number of datasets and can match or surpass alternative methods in accuracy. The timing results suggest the method is not very scalable for larger networks or with many monotonic features.

Strengths: - The proposed methods presented in the paper are novel and interesting. - The method can be used with ReLU based networks and is immediately useful in incorporating monotonicity as inductive bias. - The method makes uses off the shelf SMT solvers. - The paper is accompanied by code for the experiments.

Weaknesses: - The presented method, as is, can only be applied to networks with ReLU activations. Any activation function that is not piecewise-linear is likely not feasible to use with the methods described in the paper, as the SMT solver would need to work with a non-linear background theory. - The use for hard monotonicity guarantees requires having an SMT solver at evaluation time, which might not be plausible. - The computational cost of having a solver at evaluation time is fairly large. The method does not seem to scale well with the size of the model or when constraining more features to be monotonic.

Correctness: The presented method and theoretical contributions of the paper look sound and correct.

Clarity: The paper is clearly written and easy to follow.

Relation to Prior Work: Most of the related work either do not provide hard guarantees of monotonicity at runtime (e.g. isotonic regression), or require using a function class that can be constrained to be monotonic (e.g. lattice based models). This paper uses the more typically used ReLU network architecture, but does not extend easily to other types of activation functions. It is also not as scalable as related work.

Reproducibility: Yes

Additional Feedback: - Would be good to include a discussion on ways to extend this work to non ReLU networks. - Some rough theoretical analysis of the complexity of using the SMT solver would be nice. Judging by the timing results, I would assume an exponential complexity in the number of nodes. - Figure 3 and 4 are hard to read. Consider using a log scale for the time axis.


Review 4

Summary and Contributions: Monotonic decision making is an often seen requirement in different situations. The paper proposes counterexample guided approaches for learning neural networks that are monotonic. The paper describes two uses of counterexample-guided paradigm. One use is the iterative update of upper and lower envelopes as described in section 3.1. These upper and lower envelopes serve to ensure monotonicity. A second use is counterexample-guided learning, described in section 4.1.

Strengths: To the reviewer's best knowledge the approach is novel and very interesting. The paper reads well, but misses important detail, as summarized in the rest of the review. The experimental evaluation is very well organized. Nevertheless, the inclusion of results for a larger number of datasets would help making a stronger case.

Weaknesses: Some parts of the paper could be better written. In addition, the algorithms should include far more detail, and the presentation of the ideas should be better systematized. Additional comments are included in the additional comments box. For the sake of reproducibility, it would be important to detail the algorithms, e.g. by including the pseudo-code. If space was a problem, including the algorithms in the supplementary materials would be an option. It would be important to present results for a larger number of datasets. One ought not draw conclusions from two datasets.

Correctness: As far as I could understand, the approach is correct. It would be important to include a more detailed theoretical characterization, e.g. what is the worst-case number of counterexamples?

Clarity: The paper is well-written, although some definitions seem unnecessarily complicated. The paper should include more detail for the sake of reproducibility. More comments are included in the additional comments box.

Relation to Prior Work: With a few minor comments, as far as I can understand, the paper does a good job of relating with prior work.

Reproducibility: No

Additional Feedback: Additional comments: Some parts of the paper could be easier to read. The definition of monotonicity is unnecessarily complex. For example, there is a much simpler definition on wikipedia: https://en.wikipedia.org/wiki/Monotonic_function. The same applies to definition 2. Definition 3 should be rewritten. The reviewer believes the ReLU NN should be defined as a DAG. This should be clarified. A more comprehensive and more visible set of references could be used to highlight the success of ReLU NNs. The encoding to SMT described in page 3 misses important detail. What are the variables used? How are the summations accounted for? There are other encodings of ReLU NNs. These should at least be acknowledged. One example reference is listed below. The use of OMT may represent unnecessary overhead for the problems being solved. A simple MaxSMT (based on MaxSAT) approach might be more effective in practice. Despite being easy to read, it seems that important detail would have to be added for the work to be reproducible. What is reported seems plausible, but a straightforward implementation seems beyond reach. Reference on encoding ReLU NNs: Matteo Fischetti, Jason Jo: Deep neural networks and mixed integer linear optimization. Constraints An Int. J. 23(3): 296-309 (2018) Update after rebuttal and discussion: I have read the rebuttal. Neither the rebuttal nor the discussion changed my positive opinion on this paper.

[Author Response · NeurIPS 2020]

We would like to thank all the reviewers for their thoughtful feedback. We are happy that reviewers found our work novel, the paper well-written, and our empirical evaluation well-organized and interesting. We address your comments and questions below.

**Scalability.**

(R1, 2, 3) A primary concern is the scalability of our approach. The following reasons elucidate why the paper is interesting and state-of-the-art with its current degree of scalability.

1. Prior work [12, 53] guarantees monotonicity by restricting the hypothesis class, making it easier to scale. For example, linear models restricted to positive weights are very scalable and monotonic. However, restricting the hypothesis class leads to less expressive models, which can reduce their practical applicability. Ours is the **first work** to guarantee monotonicity for ReLU networks, without restricting the model architecture. Providing such guarantees for arbitrary ReLU networks, even for the size used in our paper, is a challenging problem.

2. Our experiments use real-world models and sizes that are state-of-the-art and have natural monotonicity constraints. Model architectures were chosen based on grid search and not based on their simplicity. From Figure 3, we can see that average monotonic prediction time for these models is less than 1 second.

3. When violating monotonicity leads to safety, ethical or legal problems, and PR disasters, the question is not whether we can scale monotonicity enforcement, but whether it is safe to use machine learning at all. In this context, the computational price of enforcing monotonicity, even if it ends up being significant, is entirely warranted.

4. Our algorithm relies on off-the-shelf SMT solvers; as there are improvements in automated solvers, our approach will directly benefit from the improvements. Future works can also investigate other efforts to improve scalability in solvers. As a concrete example, as (R4) suggested, one can compare prediction performance when using MaxSAT instead of OMT.

**Robustness.**

(R2) "This algorithm also appears to lack robustness". We respectfully disagree, in fact our approach **improves robustness**. First, we want to clarify that the monotonic envelope is constructed on the learned function and not on the data. Therefore, individual data outliers will not affect it too much. Second, if the function to be learned is naturally monotonic, enforcing invariants counteracts noise and outliers, leading to improved robustness. To illustrate the improvement, we added five outlier points to the WEIGHT feature of AUTO-MPG regression dataset. Table 1 shows that our approach produces **more robust models** with COMET improving baseline MSE. Since this is such an appealing advantage of our proposed technique, we will add these robustness experiments to the final paper.

Table 1: COMET is more robust than the baseline model ($NN_b$)

| Model | Without Outliers | With Outliers |
|---|---|---|
| $NN_b$ | 9.33±3.22 | 13.54±4.65 |
| COMET | 8.92±2.93 | 10.54±1.98 |

(R2) "Data augmentation could cause dangerous drift away from the original training data." Our approach guards against this in multiple ways. First, data augmentation with counterexamples is recomputed for each batch at every epoch. This ensures that: 1) an incorrect old counterexample does not burden the learning, and 2) learning incorporates multiple counterexamples at a time and so is less sensitive to any particular one. Second, the labeling heuristic for counterexamples (see lines 259-267 Section 4.1, page 6) provides a smoother loss with respect to monotonicity. As evidenced in our evaluation (see Table 2), there is no drift in the model quality. The quality of our model is similar or better than a model trained without monotonicity constraints.

**Other Questions and Comments.**

(R1, 3) "The presented method can only be applied to networks with ReLU activations." As noted by R2, other than the SMT encoding, the approach is **not limited** to ReLU activation functions. For other activation functions, given the encoding and an SMT solver with appropriate theory, we can directly use our algorithms. It is a non-trivial interesting future research to encode non-linear activation functions such as tanh without approximations in SMT, and we will add more discussion in the paper.

(R1, 2, 3, 4) Additional Feedback: We thank the reviewers for further suggestions. We will take these into account for improving the final draft of the paper. As suggested by (R4), we will simplify definitions and add pseudo-code to detail our algorithms further. Also, note that our code will be publicly available for reproducibility.

[Meta-Review · NeurIPS 2020]

The reviewers agreed that this was an interesting and novel approach to imposing monotonicity, and that the paper was mostly well-written (although R4’s review contains some suggested improvements). The main criticisms were that (i) the datasets in the experiments were small, and (ii) using an SMT solver at evaluation time might be too expensive for many applications. R3 also mentioned that the limitation to ReLU networks could be somewhat constraining. These issues, however, were agreed to be outweighed by the strengths of the paper, and all reviewers recommended acceptance. Please carefully read the reviews, and take them seriously when making edits: the paper is very good already, and while of course experiments should not be overhauled between submission and a final version, implementing some of the reviewers’ suggestions (especially adding a more in-depth discussion of evaluation-time costs, and their impact on real-world systems) could improve it even further.